# Long-term psychological distress trajectories and the COVID-19 pandemic in three British birth cohorts: A multi-cohort study

Darío Moreno-Agostino[1,2]*, Helen L. Fisher[2,3], Alissa Goodman[1], Stephani L. Hatch[2,4], Craig Morgan[2,5], Marcus Richards[6], Jayati Das-Munshi[2,4,7‡], George B. Ploubidis[1,2‡]

1 Centre for Longitudinal Studies, UCL Social Research Institute, University College London, London, United Kingdom, 2 ESRC Centre for Society and Mental Health, King's College London, Melbourne House, London, United Kingdom, 3 King's College London, Social, Genetic & Developmental Psychiatry Centre, Institute of Psychiatry, Psychology & Neuroscience, London, United Kingdom, 4 King's College London, Department of Psychological Medicine, Institute of Psychiatry, Psychology & Neuroscience, London, United Kingdom, 5 Health Service and Population Research, Institute of Psychiatry, Psychology & Neuroscience, King's College London, London, United Kingdom, 6 MRC Unit for Lifelong Health and Ageing at UCL, University College London, London, United Kingdom, 7 South London and Maudsley NHS Trust, London, United Kingdom

‡ These authors are joint senior authors on this work.
* d.moreno@ucl.ac.uk

**Data Availability Statement:** Deidentified data and documentation on NCDS and BCS70 are available from the UK Data Service: https://ukdataservice.ac.

## Abstract

### Background

Growing evidence suggests that population mental health outcomes have worsened since the pandemic started. The extent that these changes have altered common age-related trends in psychological distress, where distress typically rises until midlife and then falls after midlife in both sexes, is unknown. We aimed to analyse whether long-term pre-pandemic psychological distress trajectories were disrupted during the pandemic, and whether these changes have been different across cohorts and by sex.

### Methods and findings

We used data from three nationally representative birth cohorts comprising all people born in Great Britain in a single week of 1946 (National Survey of Health and Development, NSHD), 1958 (National Child Development Study, NCDS), or 1970 (British Cohort Study, BCS70). The follow-up data used spanned 39 years in NSHD (1982 to 2021), 40 years in NCDS (1981 to 2001), and 25 years in BCS70 (1996 to 2021). We used psychological distress factor scores, as measured by validated self-reported questionnaires (NSHD: Present State Examination, Psychiatric Symptoms Frequency, and 28- and 12-item versions of General Health Questionnaire; NCDS and BCS70: Malaise Inventory; all: 2-item versions of Generalized Anxiety Disorder scale and Patient Health Questionnaire). We used a multilevel growth curve modelling approach to model the trajectories of distress across cohorts and sexes and obtained estimates of the differences between the distress levels observed during the pandemic and those observed at the most recent pre-pandemic assessment and at the peak in the cohort-specific pre-pandemic distress trajectory, located at midlife. We

uk/. Deidentified data and documentation on NSHD are available from https://www.nshd.mrc.ac.uk/data.

**Funding:** This paper represents independent research part supported by the Economic and Social Research Council (ESRC) Centre for Society and Mental Health at King's College London [ES/S012567/1]. DM, HLF, SLH, CM, JD, and GBP are part supported by the ESRC Centre for Society and Mental Health at King's College London [ES/S012567/1]. SLH, CM, and JD are also supported by the National Institute for Health Research (NIHR) Biomedical Research Centre at South London and Maudsley NHS Foundation Trust and King's College London and JD is also supported by the NIHR Applied Research Collaboration South London (NIHR ARC South London) at King's College Hospital NHS Foundation Trust. MR is supported by the Medical Research Council (MRC) [MC_UU_00019/3]. The funders had no role in study design, data collection and analysis, decision to publish, or preparation of the manuscript.

**Competing interests:** The authors have declared that no competing interests exist.

**Abbreviations:** COVID-19, Coronavirus Disease 2019; DiD, difference-in-differences; FIML, Full Information Maximum Likelihood; IPW, inverse probability weighting; IRT, Item Response Theory; NCDS, National Child Development Study; NHS, National Health Service; NSHD, National Survey of Health and Development; NVQ, National Vocational Qualification; PSE, Present State Examination; PSF, Psychiatric Symptoms Frequency; SEM, Structural Equation Modelling; SMD, standardised mean difference.

further analysed whether pre-existing cohort and sex inequalities had changed with the pandemic onset using a difference-in-differences (DiD) approach. The analytic sample included 16,389 participants. By September/October 2020, distress levels had reached or exceeded the levels of the peak in the pre-pandemic life-course trajectories, with larger increases in younger cohorts (standardised mean differences [SMD] and 95% confidence intervals of $SMD_{NSHD,pre\text{-}peak} = -0.02$ [−0.07, 0.04], $SMD_{NCDS,pre\text{-}peak} = 0.05$ [0.02, 0.07], and $SMD_{BCS70,pre\text{-}peak} = 0.09$ [0.07, 0.12] for the 1946, 1958, and 1970 birth cohorts, respectively). Increases in distress were larger among women than men, widening pre-existing sex inequalities (DiD and 95% confidence intervals of $DiD_{NSHD,sex,pre\text{-}peak} = 0.17$ [0.06, 0.28], $DiD_{NCDS,sex,pre\text{-}peak} = 0.11$ [0.07, 0.16], and $DiD_{BCS70,sex,pre\text{-}peak} = 0.11$ [0.05, 0.16] when comparing sex inequalities in the pre-pandemic peak in midlife to those observed by September/October 2020). As expected in cohort designs, our study suffered from high proportions of attrition with respect to the original samples. Although we used non-response weights to restore sample representativeness to the target populations (those born in the United Kingdom in 1946, 1958, and 1970, alive and residing in the UK), results may not be generalisable to other sections within the UK population (e.g., migrants and ethnic minority groups) and countries different than the UK.

## Conclusions

Pre-existing long-term psychological distress trajectories of adults born between 1946 and 1970 were disrupted during the COVID-19 pandemic, particularly among women, who reached the highest levels ever recorded in up to 40 years of follow-up data. This may impact future trends of morbidity, disability, and mortality due to common mental health problems.

## Author summary

### Why was this study done?

- The COVID-19 pandemic has negatively impacted the mental health of the population, with disproportionate effects among specific subgroups such as women and younger people.

- Previous research suggests that, in the UK population, long-term trends of psychological distress are expected to reach their highest point during midlife (around age 30 to 45) and decrease towards older age.

- Little is known about where the potential impact of the COVID-19 pandemic stands in relation to those long-term trends of psychological distress, and whether this impact has been different across cohorts and sexes.

## What did the researchers do and find?

- We used data on 16,389 participants from three British birth cohorts representing people born in Britain in 1946, 1958, and 1970, with data on psychological distress collected between 1982 and 2021 (age 36 to 75), 1981 and 2021 (age 23 to 63), and 1996 and 2021 (age 26 to 51), respectively.

- We measured the long-term psychological distress trajectories of different cohorts (people born in 1946, 1958, and 1970) and sexes (women and men).

- We found that psychological distress levels increased during the COVID-19 pandemic, reaching or exceeding the highest levels ever recorded in up to 40 years of data, and that this increase was larger among women.

## What do these findings mean?

- This study suggests that, during the COVID-19 pandemic, there has been a new peak in the long-term trajectories of psychological distress in the UK population, one that was largely unexpected considering pre-existing trends, in addition to the peak already observed in midlife.

- This new peak in the psychological distress trajectories has been substantially larger in women than in men, widening the sex inequalities already existing prior to the pandemic onset.

- This new peak in distress may increase the trends of morbidity, disability, and mortality due to common mental health problems, with women likely being disproportionately affected. Public policies aimed at the provision of support and monitoring of population mental health, particularly among those most disproportionately affected by the pandemic, are needed to tackle existing and prevent future inequalities.

## Introduction

Mental disorders are among the leading global contributors to years lived with disability [1,2]. Growing evidence suggests that this may have worsened given the impact of the Coronavirus Disease 2019 (COVID-19) pandemic and the restriction measures put in place to control its spread, on mental health, including depression, anxiety, and, more generally, psychological distress [3–7]. In the United Kingdom, results from 11 longitudinal population-based studies show that psychological distress levels have been, overall, higher throughout the first year after the pandemic onset compared to pre-pandemic levels [8]. This complements earlier evidence focused on the initial stages of the pandemic, where worsening levels of mental health outcomes—particularly anxiety and distress levels—were reported [9–13]. Although these studies are crucial to understand whether population mental health has worsened during the pandemic, they do not provide evidence on where these changes stand in relation to pre-existing long-term mental health trajectories. In other words, how do psychological distress levels experienced during the pandemic compare to those experienced by the same individuals throughout their life course?

The answer to this question is particularly important as psychological distress levels are expected to change with age. For instance, evidence prior to the pandemic using data from three British birth cohorts (those born in 1946, 1958, and 1970) has shown that, throughout adulthood, there seems to exist an upwards trend in the long-term psychological distress trajectories by middle age (age 30 to 45), and a decrease towards older age [14,15]. Across these cohorts, the pandemic occurred at different life stages, with those born in 1970 experiencing or having recently experienced the midlife peak in distress, and those born in 1946 being further on in the decreasing trend towards older age. By extending the abovementioned life course analyses to include data collected during the first year after the COVID-19 pandemic onset, we aim (1) to understand whether the changes in distress reflect a continuation or an alteration/disruption of these pre-pandemic trends (i.e., are the changes in line with the trends observed prior to the pandemic or not?); and (2) to provide relevant insights on the magnitude of the distress levels experienced during the pandemic by comparing them not only to recent pre-pandemic levels but also to the highest levels recorded in the cohort-specific trajectory. This may have important implications for future trends of morbidity, disability, and mortality [2,16], particularly in light of the most recent results of the Global Burden of Disease Study [2], which show that, right before the pandemic onset, common mental health problems remained among the leading causes of burden worldwide. Moreover, evidence on the changes in mental health outcomes suggest that women and younger adults have been generally hit harder by the pandemic [9–13], in agreement with global evidence [17]. By analysing these long-term psychological distress trajectories across cohorts and sexes, we also aim to explore whether there are inequalities in the potential disruption of the pre-existing long-term trends across cohorts and sexes.

## Methods

### Sample and procedure

We used data from three British birth cohorts: the National Survey of Health and Development (NSHD) [18], the National Child Development Study (NCDS) [19], and the British Cohort Study (BCS70) [20], representing people born in a single week in Britain in 1946, 1958, and 1970, respectively. Life-course data from the studies were augmented with the COVID-19 Survey [21], which collected relevant information regarding the pandemic on the members of these cohort studies at three time-points: May 2020 (during the first national lockdown), September to October 2020 (between the first and second national lockdowns), and February to March 2021 (during the third national lockdown). NCDS data were further augmented with data on 1,366 participants from age 62 sweep fieldwork, which started in January 2020 and had to be paused due to the pandemic onset [22]. In this study, we focused on cohort members who took part in the COVID-19 Survey in at least one time-point. Thus, participants lost to follow-up during the COVID-19 Survey (those who were no longer alive, not living in the UK, or not participating in any of the COVID-19 Survey waves) were excluded. Data collection for the COVID-19 Survey was entirely online at the first and second time-points and was supplemented by telephone interviews at the third time-point. Response rates to the COVID-19 Survey with respect to the target population (cohort members alive and still residing in the UK) in NSHD, NCDS, and BCS70 were 31.1%, 33.5%, and 23.6% in the first wave; 39.6%, 40.7%, and 29.9% in the second wave; and 35.3%, 44.2%, and 32.5% in the third wave, respectively [21].

The authors assert that all procedures contributing to this work comply with the ethical standards of the relevant national and institutional committees on human experimentation and with the Helsinki Declaration of 1975, as revised in 2008. All procedures involving human subjects/patients were approved by the National Health Service (NHS) Research Ethics Committee. All participants provided oral informed consent.

## Measures

We used data on psychological distress collected between 1982 and 2021 (NSHD, age 36 to 75), 1981 and 2021 (NCDS, age 23 to 63), and 1996 and 2021 (BCS70, age 26 to 51). In both NCDS and BCS70, psychological distress was measured with a nine-item version of the Malaise Inventory [23,24] at all time-points, including the COVID-19 survey. Previous studies have shown that, up to the most recent pre-pandemic assessment in these two cohorts, these nine items reflected equivalently the same construct over time and across cohorts and sexes [15,25]. In NSHD, different questionnaires were used over time, both prior to and during the COVID-19 pandemic. The Present State Examination (PSE) [26] was used at age 36; the Psychiatric Symptoms Frequency (PSF; based on the PSE) [27] at age 43; and, from then onwards, two different versions of the General Health Questionnaire: the GHQ-28 at ages 50 to 69, and the GHQ-12 during the COVID-19 Survey, corresponding to ages 74 to 75 [28]. The same item harmonisation procedure implemented by McElroy and colleagues [29] was used. Following this procedure, items from the GHQ-12 questionnaire, administered during the COVID-19 Survey, were mapped to specific distressing experiences, including low mood, fatigue, tension, panic, hopelessness, health anxiety, and sleep problems. The two-item versions of the Patient Health Questionnaire (PHQ-2) [30] and the Generalized Anxiety Disorder (GAD-2) [31] questionnaires were administered during the COVID-19 survey in all cohorts in addition to their corresponding psychological distress measures. Additional information on the measures and on the harmonisation process used is available in Appendix A and Appendix B in S1 Supporting Information, respectively. Due to the wide range of different measures of psychological distress across cohorts (NSHD versus NCDS and BCS70) and within NSHD, we operationalised psychological distress as a factor score (continuous). This included all cohorts and leveraged the existence of a common set of indicators of psychological distress (PHQ-2 and GAD-2) across the three cohorts during the COVID-19 Survey waves, in addition to the cohort-specific items. The common items were used as "anchor items" to estimate a psychological distress factor and derive the corresponding factor scores across cohorts and time-points using an Item Response Theory (IRT)-based linking approach [32].

As sensitivity checks, we used additional psychological distress operationalisations, in addition to the main operationalisation as a factor score. First, we operationalised psychological distress as the number of symptoms present (discrete) at each time-point. This could be directly done in NCDS and BCS70 due to the use of the same instrument across cohorts and over time, and relied on three out of the seven previously harmonised symptoms that were present across all data collection points in NSHD due to the change in the version of the GHQ used in the COVID-19 Survey. Thus, the potential number of symptoms ranged from 0 to 9 in NCDS and BCS70, and from 0 to 3 in NSHD. Second, psychological distress was operationalised as "caseness" (binary), using each of the measurement tools' recommended thresholds (Appendix A in S1 Supporting Information). Finally, an additional factor approach was implemented in NSHD using the seven previously harmonised symptoms as indicators of a latent psychological distress factor. Further details on these additional psychological distress operationalisations are available in Appendix C in S1 Supporting Information.

Information on the cohort members' biological sex as recorded at birth was used in the interaction analyses by birth sex. Information on the highest vocational/academic qualification level achieved (harmonised into National Vocational Qualification [NVQ] levels according to the procedure laid out in Dodgeon and Parsons [33]), along with the self-reported financial situation before the COVID-19 pandemic and the self-reported general health level (both collected during the COVID-19 Survey waves), was used to provide descriptive information on the samples.

## Data analyses

**Measurement invariance/equivalence testing.** To ensure that changes in the psychological distress levels were not due to changes in the properties of the measurement tools over time and across cohorts and sexes, a measurement invariance/equivalence testing procedure was implemented using a Structural Equation Modelling (SEM) framework [34]. Evidence on measurement invariance up to the required level to perform the subsequent analyses (i.e., scalar invariance) was obtained, and further details on the procedure used, along with its results, are available in Appendix D in S1 Supporting Information.

**Derivation of factor scores.** After obtaining evidence on the invariant measurement properties of the four identical psychological distress indicators in the COVID-19 Survey waves (the GAD-2 and PHQ-2 items) (Appendix D in S1 Supporting Information), these four indicators were pooled, along with the cohort-specific psychological distress indicators. A Full Information Maximum Likelihood (FIML) estimation, corrected for the clustering induced by the longitudinal design (MLR), was used. This enabled factor scores for each time-point with at least partial information available to be obtained [35]. The same procedure was implemented in the additional sensitivity checks within NSHD, where the seven previously harmonised symptoms [14,29] were used as indicators of a latent psychological distress factor, and factor scores were derived for all time-points with at least partial information, including the COVID-19 Survey waves where four out of the seven previously harmonised symptoms were missing by design.

**Trajectories of psychological distress.** To understand whether the changes in distress reflect a continuation or an alteration/disruption of the pre-pandemic trends under the different outcome operationalisations, we used a multilevel growth curve modelling approach, using linear models for the factor scores operationalisations (continuous), Poisson models for the number of symptoms operationalisation (discrete), and logistic models for the "caseness" operationalisation (binary). To model the non-linear trajectories observed in the descriptive data, we used a piecewise approach with two main segments. The first segment covered the period from the first time-point to the last pre-pandemic assessment and corresponded to the functional form reported in the previous study for this period [14], which was quadratic (inverted U-pattern) for NSHD and cubic (U-pattern followed by a decrease or stabilisation) for BCS70. An additional polynomial term (quartic) was included in NCDS to model a slight increase in the trajectory towards the last pre-pandemic assessment. The second segment covered the period from the last pre-pandemic assessment to the study period in February/March 2021 and was defined by a polynomial curve up to the cubic term to capture the observed multifaceted change.

Unadjusted models were estimated separately for each cohort. The models were also estimated, including an interaction term between each growth parameter and birth sex, to account for inequalities in these trajectories within cohorts in line with the abovementioned evidence. The random part of all these models included the variation in the initial levels (random intercepts) but not in the change over time (random slopes) as the inclusion of this additional random effect led to convergence issues.

To answer the counterfactual question of what the distress levels would have been had the COVID-19 pandemic not occurred, models estimated with data only up to the most recent pre-pandemic assessment (2015, early 2020, and 2016 in NSHD, NCDS, and BCS70, respectively) were used to obtain projections of the distress levels in 2020 and 2021. The same models used when including the data from the COVID-19 Survey waves were not rendered useful for obtaining projections, as the polynomial terms produced unlikely predictions. Therefore, a piecewise approach with two segments was used, locating the knot at the middle point of the

pre-pandemic trajectory in order to maximise the data available to estimate each of the two segments. At least three time-points per segment were necessary to enable the estimation of non-linear trajectories in each of the segments; this is, a minimum total number of five observations, with the first to the third belonging to the first segment, and the third to the fifth belonging to the second segment. The models were estimated separately for each cohort using the main psychological distress operationalisation (cross-cohort factor score). The segments comprised years 1982, 1989, and 1999 (first segment), and 1999, 2009, and 2015 (second segment) in NSHD; years 1981, 1991, and 2000 (first segment), and 2000, 2008, and 2020 (second segment) in NCDS; and years 1996, 1999, and 2004 (first segment), and 2004, 2012, and 2016 (second segment) in BCS70. These models were used to obtain 95% confidence intervals of the mean psychological distress factor score in 2020 and 2021. These confidence intervals were plotted against those obtained from the models estimated using the complete data (this is also including data from the COVID-19 Survey waves).

**Comparison of distress levels during the pandemic with most recent and highest levels.**
To address the question of how the levels of distress experienced during the pandemic compared to both recent pre-pandemic levels and also to the highest levels recorded in the cohort-specific trajectory, we obtained the standardised mean differences (SMD) in the factor scores between the peak during the pandemic and (1) the pre-pandemic peak by midlife [14] and (2) the most recent pre-pandemic assessment. These SMDs were obtained for the three cohorts both overall and by birth sex. We then used a difference-in-differences (DiD) approach to explore whether the sex differences had changed at the pandemic peak compared to those pre-pandemic points (pre-pandemic peak and most recent pre-pandemic assessment).

There were differences within the cohorts in the probability of participating in the COVID-19 Survey waves. Women and cohort members with higher educational/vocational qualification levels were more likely to participate in the survey than men and members with lower qualification levels or no qualifications, but no significant differences were found by pre-pandemic psychological distress (more details are available in Appendix 1 of the COVID-19 Survey User Guide [21]). To account for the differential probability of participating in the COVID-19 Survey waves, and thus restore sample representativeness to the target population, all models were estimated using an inverse probability weighting (IPW) approach. The weights were generated for each of the three COVID-19 Survey waves based on personal characteristics and the history of previous participation [36]. In NSHD, these weights were combined with the corresponding design weights [18]. Additional information on the derivation of these weights and their effectiveness to restore sample representativeness and reduce bias is available in the COVID-19 Survey User Guide [21]. Missingness in pre-pandemic data collection points was assumed to be random conditional on meeting the inclusion criteria (i.e., being alive and still residing in the UK, and having participated in at least one of the COVID-19 Survey waves) at the time of the study. However, as a robustness check, we derived non-response weights for the pre-pandemic data collection points following a similar procedure as the one laid out in the COVID-19 Survey User Guide [21]. We used information on early life variables (birth sex, housing tenure and crowding, parental social class during childhood, and cognitive ability), along with the number of non-responses to previous data collection points, to predict the probability of non-response to the pre-pandemic data collection points. The resulting probabilities were used in an IPW approach to estimate the multilevel growth curve models using the main psychological distress operationalisation (cross-cohort factor score), and the results were compared to those of the main analyses.

SEM models (measurement models to test invariance/equivalence and to obtain factor scores) were estimated in Mplus version 8.6 [37]. Multilevel growth curve models were estimated in Stata MP version 17.0 [38].

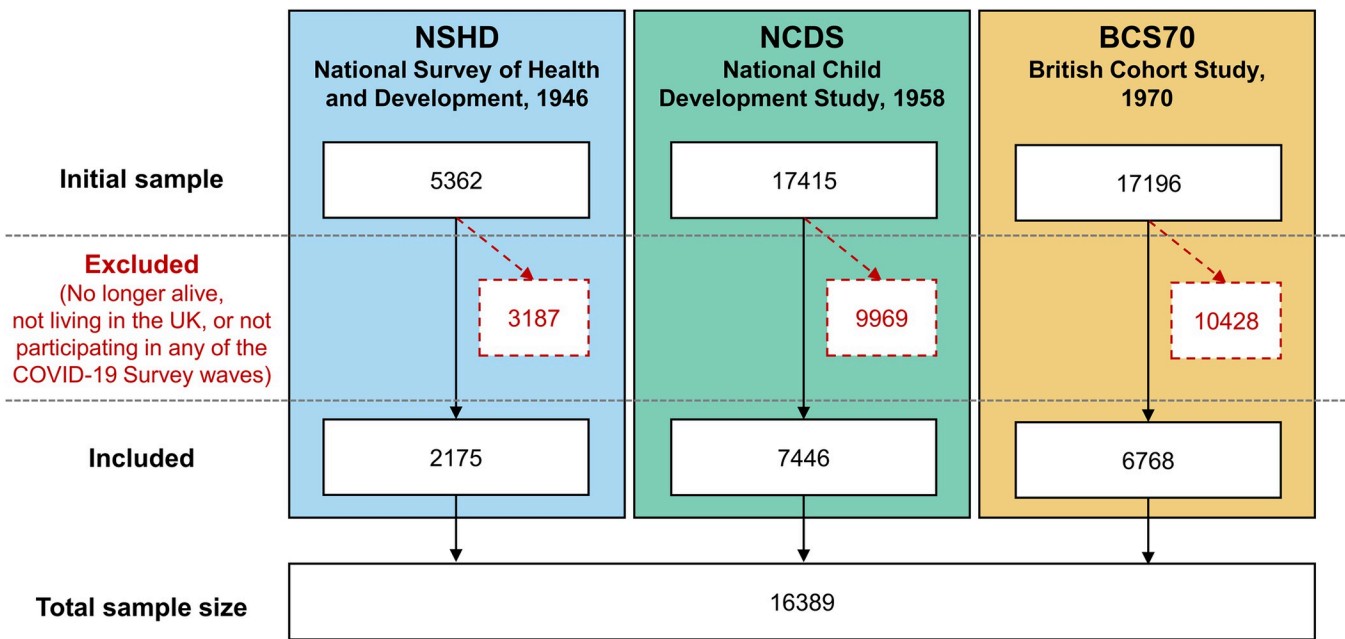

**Fig 1. Sample flow diagram.**

Analyses were planned in May 2021. The use of projections for the psychological distress levels using data up to the most recent pre-pandemic assessment, alongside SMDs and DiD estimates, were included later on as a way of supplementing and summarising the evidence of the main analyses. Robustness checks using additional non-response weights across all data collection points were included as part of the revision process.

This study is reported as per the Strengthening the Reporting of Observational Studies in Epidemiology (STROBE) guideline (Appendix E in S1 Supporting Information).

## Results

After excluding participants who did not take part in any of the COVID-19 survey waves, the overall sample comprised $N = 16,389$ participants from NSHD ($n = 2,175$, 52.8% women), NCDS ($n = 7,446$, 52.4% women), and BCS70 ($n = 6,768$, 56.2% women) (Fig 1). Members of younger cohorts had higher vocational/academic qualification levels and reported better general health levels and worse pre-pandemic financial situation than members of older cohorts (Table 1). Number of repeated observations ranged from 1 to 8 in NSHD (median = 7), NCDS (median = 6), and BCS70 (median = 6). Mean length of follow-up in the overall sample was 31.79 years (SD = 8.88), with a minimum follow-up length of 0 years (as 63 participants only had information at one time point during the COVID-19 Surveys) and a maximum follow-up length of 40 years. Cohort-specific length of follow-up was M = 37.64 (SD = 5.01, range: 0 to 39) in NSHD; M = 37.87 (SD = 5.95, range: 0 to 40) in NCDS; and M = 23.23 (SD = 4.39, range: 0 to 25) in BCS70. The number of missing observations by wave and cohort is detailed in Appendix F in S1 Supporting Information.

### Trajectories of distress as factor scores

A clear change in distress was observed in all three cohorts during the COVID-19 pandemic, which indicated a disruption to the psychological distress trajectories that had been observed

**Table 1. Sample characteristics.**

| | NSHD (N = 2,175) | NCDS (N = 7,446) | BCS (N = 6,768) |
|---|---|---|---|
| **Birth sex, N (%)** | | | |
| Male | 1,026 (47.2) | 3,541 (47.6) | 2,967 (43.8) |
| Female | 1,149 (52.8) | 3,905 (52.4) | 3,801 (56.2) |
| **Highest vocational/academic qualification level achieved, N (%)** | | | |
| None (lowest) | 633 (29.1) | 460 (6.2) | 481 (7.1) |
| NVQ-1 or equivalent | 152 (7.0) | 695 (9.3) | 431 (6.4) |
| NVQ-2 or equivalent | 445 (20.5) | 1,792 (24.1) | 1,636 (24.2) |
| NVQ-3 or equivalent | 591 (27.2) | 1,315 (17.7) | 935 (13.8) |
| NVQ-4 or equivalent | 217 (10.0) | 2,635 (35.4) | 2,302 (34.0) |
| NVQ-5 or equivalent (highest) | 20 (0.9) | 385 (5.2) | 532 (7.9) |
| *Missing* | 117 (5.4) | 164 (2.2) | 451 (6.7) |
| **Self-reported financial situation before COVID-19 pandemic onset, N (%)***  | | | |
| Just about getting by / Finding it quite difficult / Finding it very difficult | 126 (5.8) | 770 (10.3) | 1,109 (16.4) |
| Doing all right | 584 (26.9) | 2,489 (33.4) | 2,666 (39.4) |
| Living comfortably | 1,437 (66.1) | 4,017 (53.9) | 2,865 (42.3) |
| *Missing* | 28 (1.3) | 170 (2.3) | 128 (1.9) |
| **Self-reported general health level, N (%)***  | | | |
| Poor | 59 (2.7) | 291 (3.9) | 201 (3.0) |
| Fair | 313 (14.4) | 925 (12.4) | 741 (10.9) |
| Good | 797 (36.6) | 2,440 (32.8) | 2,186 (32.3) |
| Very good | 757 (34.8) | 2,876 (38.6) | 2,738 (40.5) |
| Excellent | 207 (9.5) | 885 (11.9) | 884 (13.1) |
| *Missing* | 42 (1.9) | 29 (0.4) | 18 (0.3) |

BCS70: 1970 British Cohort Study; NCDS: 1958 National Child Development Study; NVQ: harmonised (based on Dodgeon and Parsons [33]) qualification categories according to the NVQ system (higher numbers represent higher qualification); NSHD: 1946 National Survey of Health and Development.

*Self-reported information on financial situation and general health level corresponds to the earliest.

prior to the start of the pandemic across the cohorts. The unadjusted marginal predicted mean psychological distress levels (Fig 2) increased from the pandemic onset onwards and, by September/October 2020 (between first and second national lockdowns, second of the last three points in the figure), they had reached (NSHD) or exceeded (NCDS and BCS70) the highest average distress levels in the pre-pandemic trajectories. A decrease was then observed towards the last point, corresponding to February/March 2021 (during third national lockdown) in both NSHD and BCS70, whereas mean levels slightly increased further in NCDS. In all cases, distress levels by the last observation were notably higher than the last pre-pandemic levels. Models' coefficients using the cross-cohort factor score operationalisation are available in Table 2, and the resulting marginal predicted levels are available in Appendix G in S1 Supporting Information.

The psychological distress projections obtained from the models using only pre-pandemic data (Fig 3) also supported the notion of an alteration in the long-term trajectories of distress with the pandemic onset.

The interaction terms between birth sex and the parameters corresponding to the changes during the pandemic (spline 2, Table 2) were only statistically significant for NCDS ($B_{NCDS,spline2linear*women} = 0.70$ [0.32, 1.08], $p < 0.001$; $B_{NCDS,spline2quadratic*women} = -0.87$ [−1.55, −0.20], $p = 0.011$; $B_{NCDS,spline2cubic*women} = 0.33$ [0.01, 0.65], $p = 0.043$), evidencing a significantly different trajectory during the pandemic between men and women. The visual exploration of the

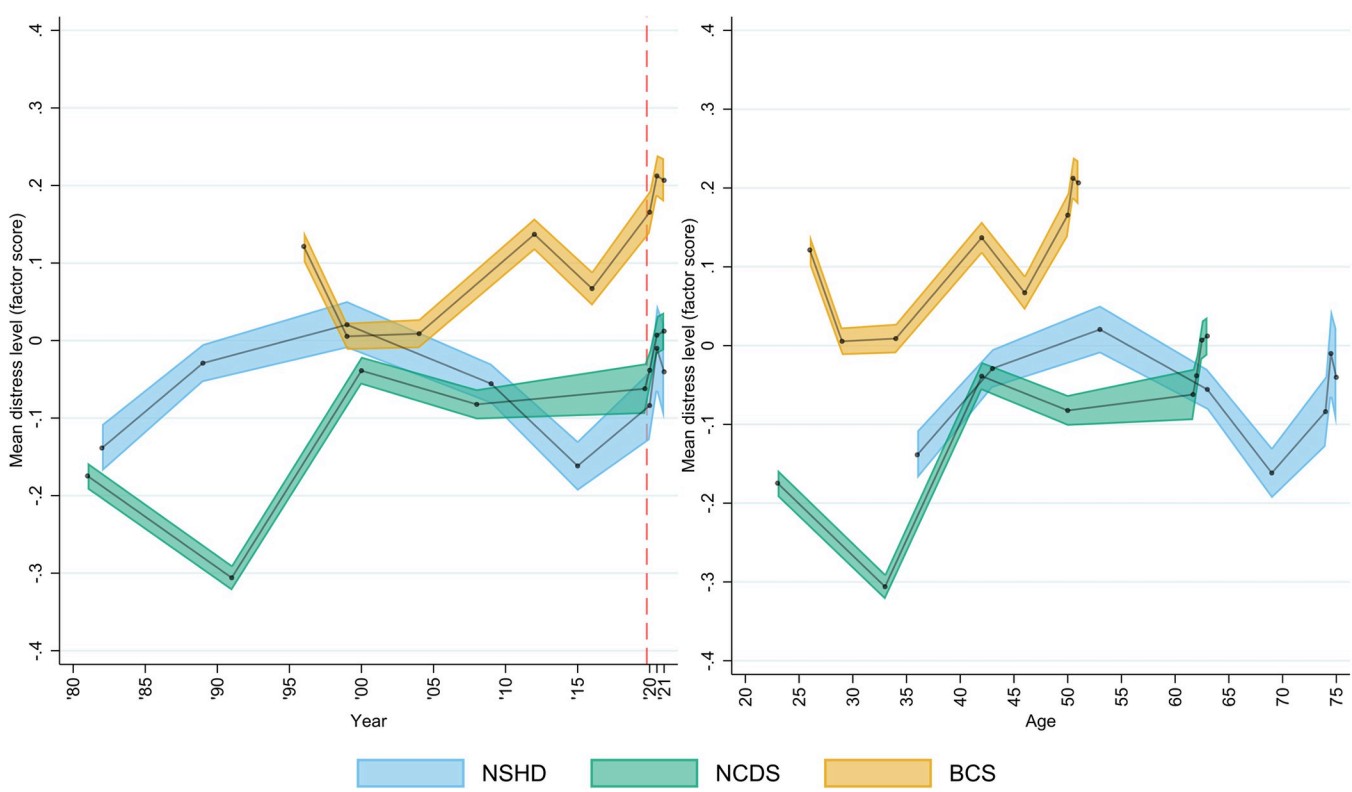

**Fig 2. Marginal mean psychological distress cross-cohort factor scores over time (year and age).** Unadjusted results. 95% confidence intervals are indicated in lighter shaded areas. BCS70: 1970 British Cohort Study; NCDS: 1958 National Child and Development Study; NSHD: 1946 National Survey of Health and Development. The dashed line represents the first nationwide lockdown enforced in March 2020.

marginal predicted levels by birth sex obtained from these models (Appendix G in S1 Supporting Information) confirmed this, showing differences in the trajectories during the pandemic across the other two cohorts as well.

## Comparison between levels during the pandemic and pre-pandemic levels

Fig 4 shows the SMD in the distress factor scores between September/October 2020 and the pre-pandemic peak in midlife (left section) and the most recent pre-pandemic assessment (right section), both overall and by birth sex. Overall, SMD were larger when compared to the most recent pre-pandemic assessment ($SMD_{NSHD,recent} = 0.14$ [0.10, 0.19], $p < 0.001$; $SMD_{NCDS,recent} = 0.05$ [0.02, 0.09], $p = 0.003$; $SMD_{BCS70,recent} = 0.14$ [0.12, 0.16], $p < 0.001$) than to the pre-pandemic peak in midlife ($SMD_{NSHD,pre-peak} = -0.02$ [−0.07, 0.04], $p = 0.518$; $SMD_{NCDS,pre-peak} = 0.05$ [0.02, 0.07], $p < 0.001$; $SMD_{BCS70,pre-peak} = 0.09$ [0.07, 0.12], $p < 0.001$), and differences with the pre-pandemic peak in midlife were larger in younger cohorts. In all cases, the overall SMD concealed the underlying sex inequalities, with women showing larger differences than men. The DiD analysis supported this observation, showing that, in all cohorts, sex inequalities had widened by September/October 2020, compared to those observed in the pre-pandemic peak in midlife ($DiD_{NSHD,sex,pre-peak} = 0.17$ [0.06, 0.28], $p = 0.002$; $DiD_{NCDS,sex,pre-peak} = 0.11$ [0.07, 0.16], $p < 0.001$; $DiD_{BCS70,sex,pre-peak} = 0.11$ [0.05, 0.16], $p < 0.001$) and in the most recent pre-pandemic assessment ($DiD_{NSHD,sex,recent} = 0.14$ [0.04, 0.24], $p = 0.005$; $DiD_{NCDS,sex,recent} = 0.15$ [0.08, 0.23], $p < 0.001$; $DiD_{BCS70,sex,recent} = 0.09$ [0.05, 0.14], $p < 0.001$).

**Table 2. Model coefficients from the multilevel growth curve models with cross-cohort factor scores as outcome (linear models).**

| | NSHD | | NCDS | | BCS70 | |
|---|---|---|---|---|---|---|
| **Models without interaction by birth sex** | **Coefficient (95% CI)** | **p** | **Coefficient (95% CI)** | **p** | **Coefficient (95% CI)** | **p** |
| Spline 1, linear term | 0.02 (0.02, 0.02) | <0.001 | −0.10 (−0.11, −0.09) | <0.001 | −0.06 (−0.07, −0.05) | <0.001 |
| Spline 1, quadratic term | −0.001 (−0.001, 0.000) | <0.001 | 0.013 (0.012, 0.014) | <0.001 | 0.007 (0.007, 0.008) | <0.001 |
| Spline 1, cubic term | | | −0.0005 (−0.0006, −0.0005) | <0.001 | −0.0002 (−0.0003, −0.0002) | <0.001 |
| Spline 1, quartic term | | | 0.00001 (0.00001, 0.00001) | <0.001 | | |
| Spline 2, linear term | −1.17 (−2.58, 0.24) | 0.105 | 0.04 (−0.15, 0.23) | 0.664 | −0.47 (−0.92, −0.02) | 0.041 |
| Spline 2, quadratic term | 0.43 (−0.09, 0.95) | 0.104 | 0.11 (−0.22, 0.45) | 0.511 | 0.22 (0.02, 0.42) | 0.031 |
| Spline 2, cubic term | −0.04 (−0.09, 0.01) | 0.111 | −0.08 (−0.24, 0.08) | 0.343 | −0.02 (−0.05, 0.00) | 0.033 |
| Intercept | −0.14 (−0.17, −0.11) | <0.001 | −0.17 (−0.19, −0.16) | <0.001 | 0.12 (0.10, 0.14) | <0.001 |
| Intercept variance | 0.20 (0.18, 0.22) | | 0.34 (0.33, 0.35) | | 0.40 (0.38, 0.41) | |
| **Models with interaction by birth sex** | **Coefficient (95% CI)** | **p** | **Coefficient (95% CI)** | **p** | **Coefficient (95% CI)** | **p** |
| Spline 1, linear term | 0.02 (0.01, 0.03) | <0.001 | −0.08 (−0.09, −0.07) | <0.001 | −0.04 (−0.05, −0.03) | <0.001 |
| Spline 1, quadratic term | −0.001 (−0.001, 0.000) | <0.001 | 0.012 (0.010, 0.013) | <0.001 | 0.006 (0.005, 0.007) | <0.001 |
| Spline 1, cubic term | | | −0.0005 (−0.0005, −0.0004) | <0.001 | −0.0002 (−0.0002, −0.0002) | <0.001 |
| Spline 1, quartic term | | | 0.00001 (0.00001, 0.00001) | <0.001 | | |
| Spline 2, linear term | −2.03 (−4.15, 0.10) | 0.062 | −0.32 (−0.58, −0.05) | 0.020 | −0.38 (−1.09, 0.33) | 0.291 |
| Spline 2, quadratic term | 0.73 (−0.06, 1.51) | 0.070 | 0.55 (0.08, 1.03) | 0.021 | 0.17 (−0.14, 0.49) | 0.279 |
| Spline 2, cubic term | −0.06 (−0.14, 0.01) | 0.078 | −0.24 (−0.47, −0.02) | 0.033 | −0.02 (−0.05, 0.02) | 0.289 |
| Intercept * women | 0.14 (0.08, 0.20) | <0.001 | 0.39 (0.36, 0.42) | <0.001 | 0.35 (0.31, 0.39) | <0.001 |
| Spline 1, linear term * women | 0.00 (−0.01, 0.01) | 0.900 | −0.03 (−0.04, −0.01) | <0.001 | −0.03 (−0.05, −0.02) | <0.001 |
| Spline 1, quadratic term * women | 0.000 (0.000, 0.000) | 0.986 | 0.002 (0.000, 0.004) | 0.052 | 0.002 (0.001, 0.004) | 0.008 |
| Spline 1, cubic term * women | | | −0.0001 (−0.0001, 0.0000) | 0.209 | −0.0001 (−0.0001, 0.0000) | 0.068 |
| Spline 1, quartic term * women | | | 0.00000 (0.00000, 0.00000) | 0.397 | | |
| Spline 2, linear term * women | 1.65 (−1.16, 4.46) | 0.249 | 0.70 (0.32, 1.08) | <0.001 | −0.15 (−1.06, 0.76) | 0.748 |
| Spline 2, quadratic term * women | −0.57 (−1.61, 0.46) | 0.279 | −0.87 (−1.55, −0.20) | 0.011 | 0.08 (−0.32, 0.48) | 0.701 |
| Spline 2, cubic term * women | 0.05 (−0.04, 0.14) | 0.301 | 0.33 (0.01, 0.65) | 0.043 | −0.01 (−0.05, 0.04) | 0.690 |
| Intercept | −0.21 (−0.25, −0.17) | <0.001 | −0.38 (−0.40, −0.36) | <0.001 | −0.08 (−0.11, −0.05) | <0.001 |
| Intercept variance | 0.19 (0.17, 0.21) | | 0.31 (0.30, 0.32) | | 0.38 (0.37, 0.39) | |

Unadjusted results. BCS70: 1970 British Cohort Study; NCDS: 1958 National Child Development Study; NSHD: 1946 National Survey of Health and Development. The intercept corresponds to the age at the first collection of psychological distress data in adulthood, being age 36 in NSHD, age 23 in NCDS, and age 26 in BCS70.

## Sensitivity checks

Analyses performed using the cross-cohort factor score operationalisation including non-response weights at all time-points (Appendix H in S1 Supporting Information), and those with the observed "number of symptoms" operationalisation (Appendix I in S1 Supporting Information), the "caseness" operationalisation (Appendix J in S1 Supporting Information), and the factor scores derived from the seven harmonised indicators within NSHD (Appendix K in S1 Supporting Information) provided very similar results as those found in the main analyses. In all these alternative operationalisations, psychological distress levels in all cohorts reached an all-time peak by September/October 2020, and a larger alteration with the pandemic onset was observed in the oldest cohort (NSHD) when using the "caseness" operationalisation.

## Discussion

Our study aimed to investigate if there had been a disruption in the pre-existing long-term psychological distress trajectories of the UK adult population during the COVID-19 pandemic

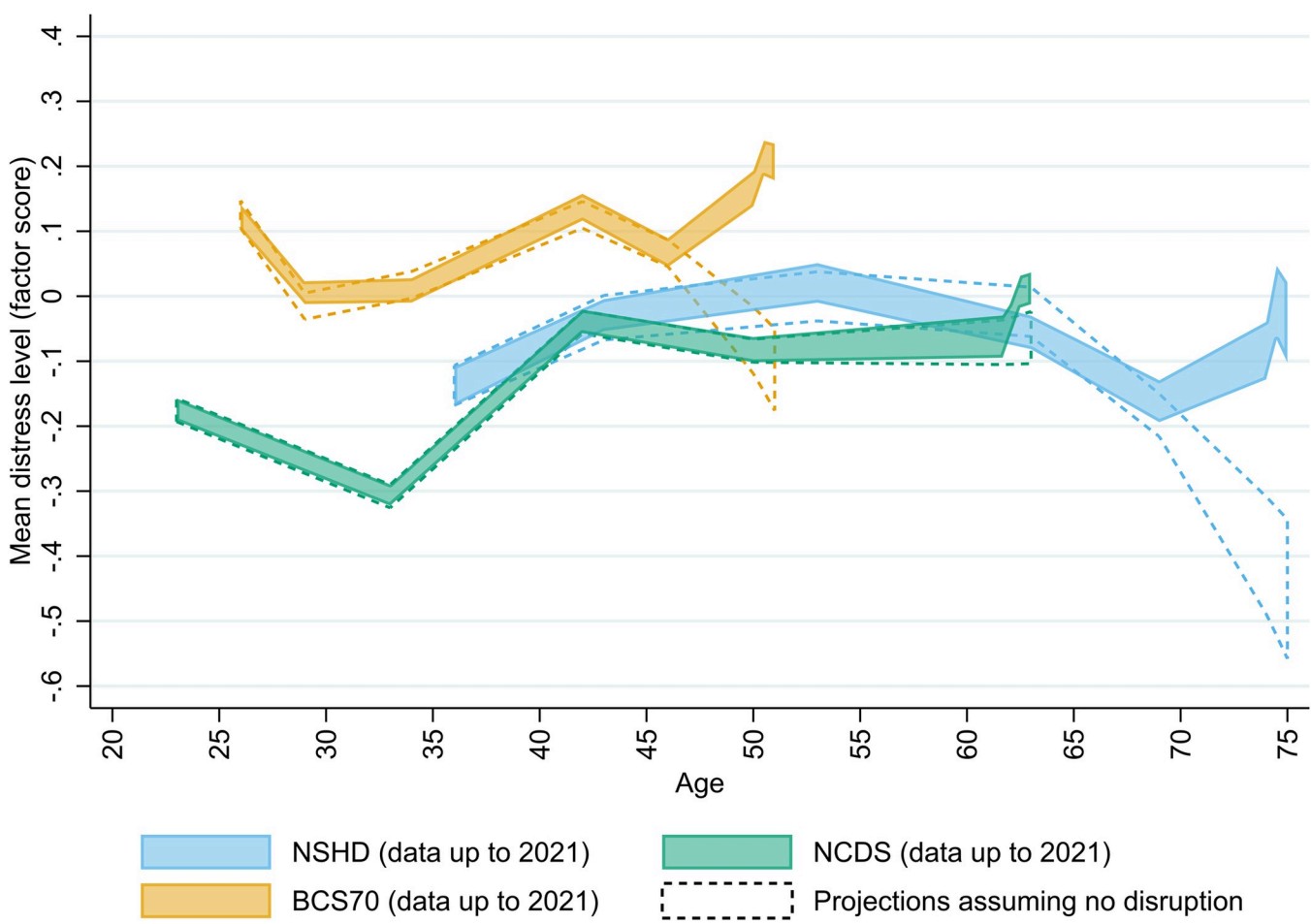

**Fig 3. Projections of mean psychological distress cross-cohort factor scores assuming no disruption.** Unadjusted results. All areas correspond to 95% confidence intervals. BCS70: 1970 British Cohort Study; NCDS: 1958 National Child and Development Study; NSHD: 1946 National Survey of Health and Development. Projections assuming no disruption are based on data up to 2015 (NSHD), 2020 (NCDS), and 2016 (BCS70).

and to analyse if such disruptions were related to the pandemic. We used a triangulation approach in the three oldest British birth cohorts, born in 1946, 1956 and 1970, using observed data on different distress operationalisations before and during the pandemic, obtaining projections based on pre-pandemic data, and examining the differences between relevant time-points before and after the pandemic onset. All these different approaches suggest that the pre-existing long-term distress trajectories, which had reached their peak by midlife (around age 40 to 50), were altered during the first year of the COVID-19 pandemic. Distress levels increased with respect to pre-pandemic levels, in most cases reaching the highest average levels over the life-course by September/October 2020. Although average distress levels tended to decrease afterwards, they were notably higher than before the pandemic onset one year after the first national lockdown. Our study also suggests that this pattern was significantly worse in women than in men regardless of age. The emergence of a new peak in the distress trajectories may increase the morbidity, disability, and mortality due to common mental health problems, which were already among the leading causes of global burden of disease without accounting for this new peak [1,2,16], with women likely being disproportionately affected by these potential increases, which may result in even greater inequalities by sex.

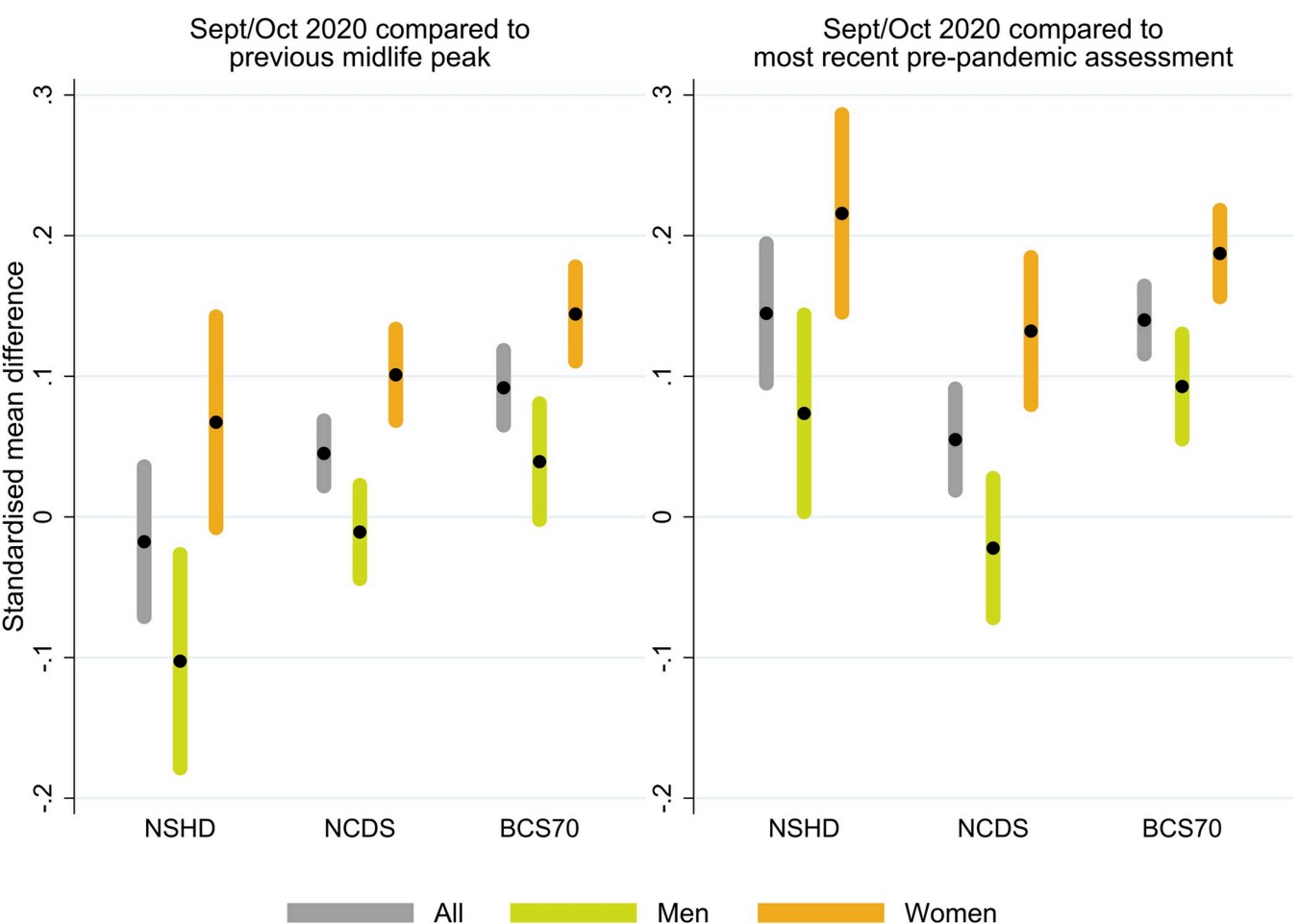

**Fig 4. Standardised mean difference in cross-cohort factor scores between September/October 2020 and pre-pandemic peak in midlife and most recent pre-pandemic assessment.** Unadjusted results. 95% confidence intervals are indicated in shaded areas. BCS70: 1970 British Cohort Study; NCDS: 1958 National Child and Development Study; NSHD: 1946 National Survey of Health and Development. Previous midlife peaks correspond to years 1999 (NSHD), 2000 (NCDS), and 2012 (BCS70). Most recent pre-pandemic assessments correspond to 2015 (NSHD), January/March 2020 (NCDS), and 2016 (BCS70).

The finding of an increase in psychological distress with regard to pre-pandemic levels is consistent with previous evidence showing an overall deterioration in mental health outcomes in the UK adult population [10–12], or in adults over the age of 50 [9,13]. The difference between the levels reached during the pandemic and the corresponding pre-pandemic peak was generally larger among younger cohorts regardless of sex. Considering that younger cohorts had higher levels of distress throughout the adulthood before the pandemic [14,15], these results may also point at future increasing inequalities by cohort. However, this finding was not consistent across the additional psychological distress operationalisations in this study. This, along with the steady levels by the last time-point in NCDS, compared to the decreasing levels observed in the other two cohorts, points at the need for further monitoring and study of these cohort inequalities.

In line with previous evidence [9–13,17], we found that women had worse distress levels than men throughout the COVID-19 pandemic, as noted. Although distress levels were already higher in women throughout adulthood, the change observed with the pandemic was larger in women. By September/October 2020, women's distress levels exceeded the levels observed in the most recent pre-pandemic assessment in all cohorts and exceeded (or reached,

while men did not) the levels observed in the pre-pandemic peak. Our study suggests that sex inequalities in psychological distress during the pandemic may not just be a continuation of pre-pandemic long-term inequalities, suggesting that these widened during the pandemic. Women have taken a disproportionately larger share of the unpaid care work responsibilities arising from pandemic control measures, including housework, homeschooling, and caring responsibilities [39,40]. Rates of domestic and gender-based violence and abuse have also reportedly increased during lockdowns [41,42]. Moreover, recent evidence suggests that, in addition to first-hand bereavements through the loss of loved ones during the pandemic, the mental health of women aged 50 and older may have also been affected by the collective, larger-scale death toll of the pandemic [43], which in the UK remains one of the highest in Europe [44]. These different factors may partly explain the larger disruption of the pre-existing long-term distress trajectories experienced by women during the pandemic.

Overall, our study suggests that the COVID-19 pandemic had a major impact on the mental health of the UK adult population. The causal mechanisms driving those adverse effects are manifold, likely including the impact and fear of the disease and the lockdown measures and subsequent limitations to the usual day-to-day activities. However, the finding that some of the worst psychological distress levels observed during the pandemic did not take place during lockdown periods suggest, in line with previous evidence [8], that the lockdown measures are not the only—or even the main—factor driving those adverse effects. Rather, the larger-scale impacts of the pandemic on the people's and country's financial situation and on other disrupted systems such as health services (crucially including mental health services [45]), may be of great importance at explaining these adverse effects and why they remained or even worsened during non-lockdown periods. The results of our study partly align with evidence from countries such as the Netherlands where, almost a year after the pandemic onset, depressive and worry symptoms remained higher than before the pandemic onset in people with no record of psychiatric disorders, whereas anxiety symptomatology gradually returned to its initial levels [46]. However, the comparison of our findings with those from different countries (even those geopolitically similar to the UK) may be difficult due to the overlap between the pandemic—with the first wave of COVID-19 and introduction of restrictions happening in March 2020 [47]—and the UK's exit from the European Union (Brexit)—with the transition period taking place for most of 2020 and the UK leaving the European Union on 1 January 2021 [48]. The role of these two events on the abovementioned financial and health services systems may be intertwined and difficult to disaggregate as they both have been happening roughly at the same time [49]. The finding that women, already disadvantaged prior to the pandemic, experienced even worse effects points in the same direction, as such inequalities are unlikely to be solely due to the differential effects of the disease and the lockdown measures by themselves. Rather, as mentioned above, the widened sex inequalities likely reflect pre-existing differences in socialisation and oppression that may have been accentuated in pandemic times [39–42]. The results of our study highlight how public policies aimed at the provision of support and continued monitoring of population mental health, particularly focused on the most disadvantaged groups (women, in our study), are very much needed to prevent further widening of inequalities. Furthermore, they serve as a warning for future lockdown-type measures to account for the differential impact of such measures in interaction with pre-existing oppression systems that may further jeopardise the mental health status of those most disadvantaged.

## Strengths and limitations

Our study has several strengths. It is, to the best of our knowledge, the longest longitudinal study of psychological distress trajectories to date, following the same individuals for up to 40

years and showing the unique effect of the pandemic over the life-course. Using data from birth cohorts enabled us to understand the potential impact of the COVID-19 pandemic in the context of the distress levels experienced by the same individuals throughout their adulthood prior to the pandemic's onset, with data collected prospectively, and a high degree of generalisability, due to the cohorts being nationally representative. Through the use of an IRT-based linking approach leveraging the existence of common distress indicators across the birth cohorts used, we were able to increase the comparability across these cohorts compared to previous evidence [14]. By using multiple operationalisations of psychological distress, including but not limited to binary outcomes, we qualify previous evidence focused on the latter [8], showing that our main results are robust to these different operationalisations while acknowledging the differences across them. Our study also has limitations. As expected in cohort designs, our study suffered from high proportions of attrition with respect to the original samples. To limit the impact of attrition, we used non-response weights, which have been found to be effective at restoring sample representativeness with respect to the characteristics of the respective target populations: those born in the UK in 1946, 1958, or 1970, alive and residing in the UK [21]. However, although this study's results may be representative of these target populations, they may not be generalisable to other sections within the UK adult population (such as migrants and ethnic minority groups, which by 2019 made up about 14% of the UK's population [50] and 15% of the population in England and Wales [51], respectively) and countries different than the UK (particularly those with different cultural, socioeconomic, and political characteristics) [52]. Finally, it was obviously not possible to include a contemporaneous control group unexposed to the pandemic in the analysis. Although we used projections based exclusively on pre-pandemic data in order to resemble the expected distress levels had the pandemic not occurred, we are aware that these counterfactual analyses have their own limitations: First, they are based on a small number of pre-pandemic data points, which limited the granularity of the predictions; second, the last time-point used in NCDS corresponded to the period just before the national lockdown came into force, and, therefore, participants may have already been preoccupied with the pandemic. This may partly explain why these projections showed a substantially smaller increase in NCDS, but further research is needed to clarify whether this was the case. It is also possible that the change observed with the pandemic was the result of pre-existing trends and unrelated to the pandemic. However, this is unlikely considering the triangulation of the results from the different analyses using data from three different cohorts, which support the notion of a pandemic-related disruption to long-term psychological distress trajectories.

## Conclusions

This longitudinal study conducted with three prospective UK birth cohorts shows that pre-existing long-term psychological distress trajectories of adults born between 1946 and 1970 were disrupted during the COVID-19 pandemic, reaching or exceeding the highest levels previously recorded in up to 40 years of follow-up data. This disruption may lead to increases in the morbidity, disability, and mortality due to common mental health problems, particularly among women, whose distress trajectories have been disproportionately altered, resulting in growing sex inequalities. Public policies aimed at the provision of support and continued monitoring of population mental health are crucial in light of these results, with a focus on those most disproportionately impacted.

## Supporting information

**S1 Supporting Information. Supporting information.** Appendix A. Information on the psychological distress measures used in the study. Appendix B. Harmonised items across

psychological distress measures in NSHD. Appendix C. Additional psychological distress operationalisations used in sensitivity checks. Figure C1. Proportion of NSHD participants endorsing each GHQ-12 item in the COVID-19 Survey waves. Table C1. Weighted prevalence estimates (and 95% CIs) of psychological distress "caseness" across COVID-19 Survey waves by operationalisation. Appendix D. Measurement invariance testing. Table D1. Malaise inventory measurement invariance testing results. Table D2. GAD-2 and PHQ-2 measurement invariance testing results. Appendix E. STROBE checklist for cohort studies. Appendix F. Number of missing observations by cohort and wave. Appendix G. Results from multilevel growth curve models with cross-cohort factor scores as outcome (linear models). Table G1. Marginal mean levels predicted from the multilevel growth curve models with cross-cohort factor scores as outcome (linear models). Figure G1. Marginal mean cross-cohort psychological distress factor scores over time (year) by birth sex. Appendix H. Results from multilevel growth curve models with cross-cohort factor scores as outcome (linear models), sensitivity checks with extended non-response weights. Table H1. Model coefficients from the multilevel growth curve models with cross-cohort factor scores as outcome (linear models), sensitivity checks with extended non-response weights. Table H2. Marginal mean levels predicted from the multilevel growth curve models with cross-cohort factor scores as outcome (linear models), sensitivity checks with extended non-response weights. Appendix I. Results from multilevel growth curve models with number of symptoms as outcome (Poisson models). Table I1. Model coefficients from the multilevel growth curve models with number of symptoms as outcome (Poisson models). Table I2. Marginal mean levels predicted from the multilevel growth curve models with number of symptoms as outcome (Poisson models). Figure I1. Marginal mean number of psychological distress symptoms over time (year and age). Appendix J. Results from multilevel growth curve models with caseness as outcome (logistic models). Table J1. Model coefficients from the multilevel growth curve models with caseness as outcome (logistic models). Table J2. Marginal mean levels predicted from the multilevel growth curve models with caseness as outcome (logistic models). Figure J1. Marginal predicted mean probability of psychological distress over time (year and age). Appendix K. Results from multilevel growth curve models with factor scores as outcome (linear models), 7 harmonised psychological distress indicators in NSHD. Table K1. Model coefficients from the multilevel growth curve models with factor scores as outcome (linear models), 7 harmonised indicators in NSHD. Table K2. Marginal mean levels predicted from the multilevel growth curve models with factor scores as outcome (linear models), 7 harmonised indicators in NSHD. Figure K1. Marginal mean psychological distress factor scores over time (year), 7 harmonised psychological distress indicators in NSHD.
(PDF)

## Acknowledgments

The British Cohort Study 1970 and National Child Development Study 1958 are supported by the Centre for Longitudinal Studies, Resource Centre 2015–20 grant [ES/M001660/1] and a host of other co-funders. The NSHD cohort is hosted by the MRC Unit for Lifelong Health and Ageing at UCL funded by the MRC [MC_UU_00019/1Theme 1: Cohorts and Data Collection]. The COVID-19 data collections in these five cohorts were funded by the UKRI grant Understanding the economic, social and health impacts of COVID-19 using lifetime data: evidence from 5 nationally representative UK cohorts [ES/V012789/1].

We would like to thank all individuals who participated in the three birth cohort studies for so generously giving up their time over so many years, and all the study team members for their tremendous efforts in collecting and managing the data. The authors would also like to

thank Dr Dawid Gondek for providing very helpful code for some of the data management and statistical analyses used in this study.

The views expressed are those of the authors and not necessarily those of the ESRC, NIHR, the Department of Health and Social Care, MRC, or King's College London.

## Author Contributions

**Conceptualization:** Darío Moreno-Agostino, Jayati Das-Munshi, George B. Ploubidis.

**Data curation:** Darío Moreno-Agostino.

**Formal analysis:** Darío Moreno-Agostino.

**Methodology:** Darío Moreno-Agostino.

**Supervision:** Jayati Das-Munshi, George B. Ploubidis.

**Visualization:** Darío Moreno-Agostino.

**Writing – original draft:** Darío Moreno-Agostino.

**Writing – review & editing:** Darío Moreno-Agostino, Helen L. Fisher, Alissa Goodman, Stephani L. Hatch, Craig Morgan, Marcus Richards, Jayati Das-Munshi, George B. Ploubidis.

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
