## [Editor Report · Decision Letter 0]

20 Jun 2022

Dear Dr Moreno-Agostino, 

Thank you for reaching out with regards to your manuscript entitled "Disruption of long-term psychological distress trajectories during the COVID-19 pandemic: evidence from three British birth cohorts".

Your manuscript has now been evaluated by the guest editors of the upcoming special issue on the pandemic and global mental health. I am writing to let you know that we would like to rescind the original decision and send your submission out for external peer review.

Please re-submit your manuscript within two working days, i.e. by Jun 22 2022 11:59PM.

Kind regards,

Callam Davidson

Associate Editor

PLOS Medicine

---

## [Decision Letter · Decision Letter 1]

3 Aug 2022

Dear Dr. Moreno-Agostino,

Thank you very much for submitting your manuscript "Disruption of long-term psychological distress trajectories during the COVID-19 pandemic: evidence from three British birth cohorts" (PMEDICINE-D-22-01955R1) for consideration at PLOS Medicine. 

Your paper was evaluated by an associate editor and discussed among all the editors here. It was also discussed with an academic editor with relevant expertise, and sent to independent reviewers, including a statistical reviewer. The reviews are appended at the bottom of this email and any accompanying reviewer attachments can be seen via the link below:

[LINK]

In light of these reviews, I am afraid that we will not be able to accept the manuscript for publication in the journal in its current form, but we would like to consider a revised version that addresses the reviewers' and editors' comments. Obviously we cannot make any decision about publication until we have seen the revised manuscript and your response, and we plan to seek re-review by one or more of the reviewers. 

We hope to receive your revised manuscript by Aug 24 2022 11:59PM. Please email us (plosmedicine@plos.org) if you have any questions or concerns.

We look forward to receiving your revised manuscript. 

Sincerely,

Callam Davidson, 

PLOS Medicine

plosmedicine.org

Please revise your title according to PLOS Medicine's style. Your title must be nondeclarative and should include the study design ("A cohort study”) in the subtitle. I would suggest “Associations between long-term psychological distress trajectories and the COVID-19 pandemic in the UK: a cohort study”, or similar. 

Please structure your abstract using the PLOS Medicine headings (Background, Methods and Findings, Conclusions).

Abstract Methods and Findings:

* Please include the years during which the study took place, length of follow up, and main outcome measures.

* Please quantify the main results (with 95% CIs and p values).

* Please include the important dependent variables that are adjusted for in the analyses.

Please include line numbering throughout your manuscript to facilitate future reviews.

Citations should be in square brackets, and preceding punctuation.

You may use almost any description as the item name of your supporting information as long as it contains an "S" and number. For example, “S1 Appendix” and “S2 Appendix,” “S1 Table” and “S2 Table,” and so forth (as opposed to ‘eAppendix #). Please see https://journals.plos.org/plosmedicine/s/supporting-information

Please specify whether informed consent was written or oral.

Please remove the ‘Role of the funding source’ section.

Please ensure that the study is reported according to the STROBE guideline, and include the completed STROBE checklist as Supporting Information. Please add the following statement, or similar, to the Methods: "This study is reported as per the Strengthening the Reporting of Observational Studies in Epidemiology (STROBE) guideline (S1 Checklist)."

Did your study have a prospective protocol or analysis plan? Please state this (either way) early in the Methods section.

Please define "lost to follow-up" as used in this study. Other reasons for exclusion should be defined.

Please define the length of follow up (eg, in mean, SD, and range), overall and by cohort. 

Please include the date of access for references 43 and 46. 

Please remove the Declarations of interest, Author contributions and Data Availability section from the main text as this information is already captured as part of the submission form. 

Similar to the above, please remove any funding information in your acknowledgements that is already covered in your Financial Disclosure (submission form).

References: Journal name abbreviations should be those found in the National Center for Biotechnology Information (NCBI) databases. 

Please use et al. after listing the first six authors in your Supporting Information references. 

Comments from the reviewers:

Reviewer #1: This is a well-executed work. My worry was that the measures of mental health collected were rather different, however the authors have used a factor score to operationalise physichological distress and

also have undertaken sensitivity analyses.

Their results showed significant distruptions during the covid-19 pandemic which in turn might have long term effects on other health outcomes.

I would have liked to see a bit more on the policy implications of this work in the discussion, especially in the context of the UK.

Reviewer #2: Thanks for the opportunity to review your manuscript. My role is as a statistical reviewer, so my review concentrates on the study design, data, and analysis that are presented. I have put general questions first, followed by queries relevant to a specific section of the manuscript (with a page/paragraph reference).

 This study characterises changes in psychological distress, during the first phase of COVID-19 in the UK using data from three ongoing birth cohort studies supplemented with additional surveys collected more intensively in 2020/2021. Data from these studies was harmonised to create a distress measure that was available for each of the cohorts at each follow-up time point. A multilevel growth model was used to characterise the trajectories of distress. This was completed with the additional 20/21 data, and without this data to estimate a counterfactual growth curve for the effect of the Covid period. Non-response to data collected was accounted for by IPW based on characteristics of participants. 

Reading this manuscript is a great reminder of the value of ongoing cohort studies - it would be nearly impossible to track long and short term changes in health without a study design like this. Also a thank-you for the well organised supplementary material - this anticipated most of my questions I had from the main manuscript. Much attention has been spent (well spent) on measurement and operationalisation of psychological distress and the approach described is reasonable. A measurement invariance approach (implemented through SEM) was used to check if changes across time could be due to changes in underlying measures. I agree with the interpretation of the different fit indices (RMSEA vs. CFI/TLI) and there is good evidence provided that scalar invariances holds for this data. Several sensitivity analyses are included, examining different approaches to the measurement of distress, and the results are very similar. 

There is really only one minor change:

P4, Paragraph 2. "we aims to understand", should be "we aim to understand"

eFig 3.1 Not a comment that needs a response - but I found the change in measure 7 very interesting. Public health is more than just health protection, and wellbeing is a vital part of health

Reviewer #3: Review: Disruption of long-term psychological distress trajectories during the COVID-19 pandemic: evidence from three British Cohorts. 

The authors investigate whether psychological distress trajectories have been disrupted by the COVID-19-pandemic using data from three large representative cohort studies in the UK. The manuscript additionally describes whether such disruptions vary by age and sex groups. The authors find that distress levels during the COVID-19 pandemic reach or exceed pre-pandemic distress levels. Increases were larger among younger and female individuals.

Overall the manuscript is sets forth interesting questions, uses an impressive dataset and presents thorough analysis that looks at the data from many different angles. However with that said, it sometimes gets confusing for the reader to make sense of the complexity and how certain choices relate to the main aims of the paper. Therefore, the manuscript could stand to benefit from some more explanation/clarification of key premises. In addition, the choice to "hide" a some relevant information in the appendices, does not enhance the overall readability of the paper. Given that the journal does not have strict restrictions on word count, I would like to invite the authors to reconsider some of these choices. Below are major and minor points in detail.

Major points:

- While the authors depart from an interesting premise that mental distress caused COVID-19 pandemic should be viewed in the context of life course trajectories of mental distress, I am not totally convinced by the arguments provided:

o Sentence: "they do not provide evidence on where these changes stand in relation to pre-existing long-term mental health trajectories" (p 4) is unclear. Changes in which direction? And what is meant by pre-existing trajectories? Can we assume that life course trajectories in previous generations form a template for how future generations will experience mental distress? Please clarify this. 

o It is unclear how the authors define a "disruption/alteration". Is a disruption defined a temporary elevation in psychological distress during the pandemic? If so, how does this add to the current literature? (as we already know that distress levels generally increases during the pandemic). Or is a disruption an elevation in distress symptoms compared to the highest distress levels measured in an individual's life course? If so what exactly does this tell us about how the life course trajectory of physical distress is altered because of the pandemic? (one suggestion to maybe make this more clear is to consider adding an graph displaying an hypothesized trajectory of the disruption in mental distress see for example Norris, F. H., Tracy, M., & Galea, S. (2009). Looking for resilience: Understanding the longitudinal trajectories of responses to stress. Social science & medicine, 68(12), 2190-2198.)

o It is unclear to me if the authors are interested in period, age or birth cohort differences throughout the manuscript. In part this is due to the fact that various terminology is used throughout. Please consider sticking with one term and formulate clearer expectations with regards to how disruptions are expected to differ depending in what life stage an individual experiences the COVID-19 pandemic (mid-life or after mid-life). Note if the aim is to investigate birth cohort differences, I would expect a broader analysis of how cohorts differ between each other beyond the age at which they experienced the pandemic (i.e. increased levels of education, socio-political changes, changed in sex inequalities across birth cohorts in general).

- Some results/analysis that seem integral to answering the research question and aims are "hidden" in the appendix. This overall does not improve the readability of the paper. Moreover, it is not always clear to me where specific results can be found, which results are coming from which analysis and which results are connected to which aim. Overall, I have a few suggestions that might make the connection between the appendix and the text more clear.

o eAppendix 1 is mentioned in the "sample and procedures" section on page 5. However, it is not mentioned what the content of this appendix is and how it adds to the description given on this page.

o Consider reporting eAppendix 6 in the methods section as this is an integral part of the research aim.

o Consider reporting eAppendix 7 in the sample description part of the methods section. 

o Consider reporting eAppendix 8 in the methods section.

o I think it would greatly enhance readability if conclusions of the analysis found within the appendix are mentioned in the text and not just where you can find the information. For example, on page 6 you mention that further details on the measurement invariance testing procedure can be found in eAppendix 4. However you do not mention whether various psychological distress levels were invariant across cohorts. This creates an unnecessary cliff-hanger. Please consider mentioning the conclusions of appendices in the text. 

- Overall the method section could use more structure in terms of which analysis/results section focuses on which aim. The same holds true for the result section. For example, in section titled "trajectories of psychological distress" it is not clear why the analysis discussed at the end of the page (SMD) was done and which research aim it was supposed to answer to. In addition it was unclear how the two relevant pre-pandemic time-points were determined. Another example, is in the beginning of the result section where it is not clear where I do not find it clear the result "A period effect was observed …across the cohorts" (p8) can be found and which aim it related to. Please clarify this. 

- While the results cover a representative population in the UK, it would be interesting if you could reflect on the meaning of these results for other countries that have experienced lockdowns. For example, in studies conducted in the Netherlands (e.g. Kok AA, Pan K-Y, Ottenheim NR, Jörg F, Eikelenboom M, Horsfall M, et al. Mental health and perceived impact during the first Covid-19 pandemic year: A longitudinal study in Dutch case-control cohorts of persons with and without depressive, anxiety, and obsessive-compulsive disorders. Journal of affective disorders. 2022) showed that mental distress levels gradually return to its initial levels, especially with regards to depression and anxiety. Does this mean that the UK potentially stands out in this regard? If so why?

minor points:

Abstract:

- Second sentence: There is an illogical comparison. "distress typically rises until mid-life and then falls [after mid-life] in both sexes." Please add "after mid-life"

- Methods: please mention the procedures/analytic method in the abstract

Introduction:

- "we aims to understand (..)" (p 4) . The s after aim should be removed.

Methods:

- How was the survey administered - online?

- What was the response rate?

- Please consider "choice of primary measure" changing into "mental distress measure"

- Please explain which "outcome operationalisations" are used in the multilevel growth curve modelling in page 7.

- How many missings where there in the pre-COVID data and how were they handled?

Results

- Consider adding a Table depicting descriptive statistics in the beginning of the result section. (particularly would it be possible to show whether the three samples differed with regards to other background characteristics: educational level, income, overall health)

Overall:

- The variable use of birth cohort, age, generation and period difference is confusing. Consider sticking with the term birth-cohort instead of generation throughout the paper.

Reviewer #4: Thank you for the opportunity to review this manuscript. The authors have attempted to estimate the likely psychological distress across 3 cohorts in the absence of the pandemic and compare this to what was observed during the pandemic. After reading this paper, I believe their attempt is decent. I was impressed by the number of statistical assumptions and sensitivity checks the authors had conducted to ensure the validity of their assumptions. They tested the approach using 3 different methods and the overall conclusions converged. Their eAppendix is well organised and detailed. While I am generally statistically inclined, I have no prior experience with the authors' IRT-based linking approach. This is probably best evaluated by someone with specific experience and knowledge with this approach. 

Page 6 - "The item harmonisation procedure reported elsewhere(14, 29) was implemented where items from these different questionnaires were mapped to specific distressing experiences." I would like the authors to make this a bit clearer here about what was done without relying on citations. Even just a sentence summary would suffice. What do the authors mean exactly by 'mapped to specific distressing experiences'? Do they mean symptoms? eAppendix 2 makes sense but nothing here mentioned about the process. 

Page 6 - "Information on the cohort members' biological sex as recorded at birth was used." used for what? Used how?

eAppendix 5 - this SEM diagram does not really make sense to me. What are the dotted boxes vs the solid boxes? Are they latent constructs of the scales? How was an SEM model run to associate or path one to another when usually measure-type completed is mutually exclusive by survey? I have never seen an SEM model illustrated like this. 

I appreciate the authors used non-response weights but did the authors investigate whether there was any non-response bias during the COVID surveys? Any significant pre-pandemic differences by demographics or prior psychological distress?

[LINK]

---

## [Decision Letter · Decision Letter 2]

2 Nov 2022

Dear Dr. Moreno-Agostino,

Thank you very much for re-submitting your manuscript "Long-term psychological distress trajectories and the COVID-19 pandemic in three British birth cohorts: a multi-cohort study" (PMEDICINE-D-22-01955R2) for review by PLOS Medicine.

I have discussed the paper with my colleagues and the academic editor and it was also seen again by three reviewers. I am pleased to say that provided the remaining editorial and production issues are dealt with we are planning to accept the paper for publication in the journal.

[LINK]

We look forward to receiving the revised manuscript by Nov 09 2022 11:59PM.   

Sincerely,

Callam Davidson, 

Associate Editor 

PLOS Medicine

plosmedicine.org

Requests from Editors:

Please ensure that all numbers presented in the abstract are present and identical to numbers presented in the main manuscript text.

Lines 418-421: I would propose that this addition requires some elaboration and should be supported by references if possible. 

Comments from Reviewers:

Reviewer #2: Thanks for the revised manuscript and replies to my original queries. The manuscript is fine from my perspective with this revision.

Reviewer 3 raises a good point about distinguishing age/period/cohort , the changes made to the manuscript make the distinction much clearer. The addition of the sensitivity test with IPW is also useful and agree with the authors there's effectively no difference between the main model and the sensitivity analysis. 

Reviewer #3: Many thanks for the opportunity to review this paper. I also thank the authors for making the many of the requested changes and for their thoughtful responses to my comments. For me the manuscript has improved greatly and only a few minor points remain: 

Author summary:

- Under the heading "what did the researchers do and find" the statement "as early as 1981" seems rather unclear. Given the emphasis of changes over de life course especially during midlife, I think it may be important to mention the age ranges and cohort memberships (in terms of birth year) of respondents included in the study in the author summary. Please consider adding this information.

- In the "what do the findings mean?" section authors may consider more clearly stating that the word "new" before peak refers to the fact that this peak was found in addition to the peak already observed in midlife (e.g.: additional peak, second peak, in addition to the peak in midlife, we now also find…).

Methods:

- It is not clear to me how old participants were at the first measurement included in the study (i.e. various figures suggest NHD: 35, NCDS: between 25-30, BCS: 20-25). Please consider adding baseline age for each cohort to Table 1

- It is not clear to me what "age" means in table S5, Is this the average age of respondents who dropped out? What would be more informative perhaps is the average age of respondents who dropped out of the study compared to the average age of respondents included in the study. 

Overall:

- In response to comment 3 by reviewer 3 the authors mention that their paper aimed to "explore and describe" differences between cohorts. However, this is not reflected in the wording of the aim where the authors refer to "investigate" difference. Please consider using wording such as "explore/describe" in order to stay closer to the what was mentioned in the response.

Reviewer #4: Thank you again for the opportunity to review this manuscript. The authors have addressed my comments and I think this is a very good piece of work. My only additional query was whether the authors had explored whether there was a sex X pre-pandemic psychological distress interaction on missingness. Even if one is found it would not be critical, just interesting to report. Non-response seems to becoming a larger issue with time and so the more information on this in the public domain the better.

[LINK]

---

## [Editor Report · Decision Letter 3]

21 Nov 2022

Dear Dr Moreno-Agostino, 

On behalf of my colleagues and the Academic Editor, Dr Lola Kola, I am pleased to inform you that we have agreed to publish your manuscript "Long-term psychological distress trajectories and the COVID-19 pandemic in three British birth cohorts: a multi-cohort study" (PMEDICINE-D-22-01955R3) in PLOS Medicine.

PRESS

PLOS frequently collaborates with press offices. If your institution or institutions have a press office, please notify them about your upcoming paper at this point, to enable them to help maximise its impact. If the press office is planning to promote your findings, PLOS would be grateful if they could coordinate with medicinepress@plos.org. As this manuscript is to be published as part of the upcoming Special Issue on the pandemic and global mental health, it will be opted out of the early version process. If you would like to discuss this further or if you have any further questions or concerns, please reach out directly (cdavidson@plos.org).

Sincerely, 

Callam Davidson 

Associate Editor 

PLOS Medicine